# Isolation and Identification of Chemical Compounds from *Agaricus blazei* Murrill and Their In Vitro Antifungal Activities

**DOI:** 10.3390/molecules28217321

**Published:** 2023-10-28

**Authors:** Ruirui Yu, Xiaojian Li, Peng Yi, Ping Wen, Shuhong Wang, Chenghui Liao, Xun Song, Haiqiang Wu, Zhendan He, Chenyang Li

**Affiliations:** 1School of Pharmacy, Shenzhen University Medical School, Shenzhen University, Shenzhen 518055, China; 15220201716@163.com (R.Y.); 15115433780@163.com (P.Y.); wuhq@szu.edu.cn (H.W.); 2College of Pharmacy, Shenzhen Technology University, Shenzhen 518118, China; xsong@szu.edu.cn; 3Shenzhen Institute for Drug Control, Shenzhen 518057, China; lanhair@126.com (P.W.);; 4School of Pharmacy, Harbin Medical University, Harbin 150081, China; liao.chenghui66@gmail.com

**Keywords:** *Agaricus blazei* Murrill, fungus, structure identification, antifungal activity

## Abstract

This study explores the antifungal properties of *Agaricus blazei* Murrill, a valuable medicinal and edible fungus. Six compounds (**1**–**6**) were first isolated from *A. blazei* using various isolation techniques and identified using spectroscopic methods. These compounds include linoleic acid, 1,1′-oxybis(2,4-di-tert-butylbenzene), glycerol monolinoleate, volemolide (17R)-17-methylincisterol, (24s)-ergosta-7-en-3-ol, and dibutyl phthalate. This study also assesses the antifungal activities of these compounds against *Trichophyton mentagrophology*, *Trichophyton rubrum*, *Candida albicans*, and *Cryptococcus neoformans*. The results demonstrate varied sensitivities against these pathogenic fungi, with compound **2** showing significant inhibition against *T. mentagrophology*, compound **3** showing significant inhibition against *T. rubrum*, and compound **6** showing significant inhibition against *C. albicans*. This study underscores the medicinal potential of *A. blazei* as an antifungal agent and sheds light on its valuable research implications.

## 1. Introduction

Fungi stand out as exceptional creators of a broad spectrum of natural products, contributing to biodiversity and offering a treasure trove of compounds imbued with a wide range of biological functions [1,2]. Notably, mushrooms, a type of fungi [3], assume a pivotal role in traditional agriculture, cuisine, and medicinal practices [4,5,6]. The fungi within these mushrooms produce unique bioactive compounds [7,8,9], with derived compounds also fulfilling nutritional roles [10]. These compounds not only serve as food sources, but also hold potential as biopharmaceuticals [11], earning them the designation of “drug alternatives”.

As we know it, there is a large number and a wide variety of fungal diseases in the world that have a great range of influence. There are several classes of antifungal drugs, including azoles, echinocandins, polyenes, and allylamines. These drugs work through various mechanisms to inhibit fungal growth and replication. Azoles, for example, interfere with fungal cell membrane formation, while echinocandins target cell wall synthesis [12,13]. Although many drugs are used as current treatment of fungal diseases, there are still many defects and deficiencies. Therefore, the development of antifungal drugs and the search for antifungal active ingredients are still of great significance. Many reports have shown that fungal families have a variety of biological activities. Therefore, the focus of this study was on the extraction, isolation, and identification of antifungal activities of ABM.

*A. blazei* (ABM), also known as the Brazilian mushroom, is native to the southern regions of North America, Brazil, Peru, and various parts of China (Figure 1). In China, it primarily thrives in high-altitude forests of northeastern regions, as well as Yunnan, Guizhou, and the Sichuan border areas. *Shennong Baicao Classic*, an ancient Chinese herbal text, documents the notable effects of *A. blazei*, such as “harmonizing the spleen and stomach” and “stabilizing the mind.” It is especially recommended for individuals with low immunity and malnutrition. Its nutritional value is substantial as it is rich in various components, including polysaccharides (such as oligosaccharides, mannan, xylose, and α- and β-glucans) [14,15,16], proteins [17], terpenes [18], fats, saponins, sterols [19,20], trace elements [21], and essential amino acids. This abundance of nutritional content underscores its significance in both cuisine and medicine [22]. Its range of biological activities is diverse, and its impact is particularly notable in various domains; for example, its anti-tumor effects [23,24,25], antiviral properties [26,27], antioxidation capabilities [16], hypoglycemic effects [28], hepatoprotective benefits [29,30], immune regulation [31], and other aspects are particularly remarkable [32]. It is considered a precious dual-purpose fungus for drugs and food [33], and it is also gaining reputation for its excellent medical value and biochemical properties. It was included in the “Redlist of China’s Biodiversity—Macrofungi” in 2018.

Professor Wu Yiyuan of the Chinese Institute of Medical Cancer showed that the tumor-inhibitory effect of agaricum Antellae, which is a close relative of *Ganoderma lucidum*, can be about five times higher than *G. lucidum*, ranking first among the various medicinal fungi; this fungus is recognized as one of the 15 effective anti-cancer medicinal fungi, and is also known as “the last food for cancer patients on Earth”. In short, ABM not only can be used as a valuable edible fungus with a beautiful taste and benefits for health care, but it also has immeasurable development potential in its scientific research and application value. However, as far as the current literature reports are concerned, most studies on ABM have focused on the anti-tumor and immune-regulating effects of its extracts [34,35] and polysaccharides [36], and there are no reports on the chemical components of non-polysaccharides and their antifungal activities. In this study, the non-polysaccharide components of ABM Antler were extracted and separated, the structure of these compounds was characterized, and their antifungal activities were studied.

Many scholars have tried to extract the chemical components of ABM, among which the effective methods [37] are acid extraction, hot water extraction, enzyme extraction, chromatography, and ethanol precipitation [38]. In the present study, a large scale extraction of *A. blazei* was carried out with reference to the above mentioned methods, from which the compounds were isolated, their structures were resolved using spectroscopic methods and their antifungal activities were evaluated. In a word, the main objective of the present work was to determine the antifungal activity of *A. blazei.*

## 2. Results

### 2.1. Identification of Compounds **1**–**6**

Linoleic acid (1): Compound **1** was extracted as a yellow oil, and its molecular formula is C_18_H_32_O_2_. This compound shows the following spectroscopic properties: ^1^H NMR and ^13^C NMR data (Appendix A) showed that the structural type of compound **1** is fatty acid. ^13^C NMR showed that there is an obvious carbonyl carbon signal (*δ_C_* 173.27), four olefinic carbons (*δ_C_* 130.24, 130.02, 128.09, 127.91) and one methyl carbon (*δ_H_* 14.08). The remaining signals are methylene carbon signals. All the obtained data of compound **1** are identical to rhapontigenin [39]. Hence, compound **1** was elucidated as linoleic acid (Figure 2).

1,1′-Oxybis(2,4-di-tert-butylbenzene) (**2**): Compound **2** was extracted as a yellowish oily matter, and its molecular formula is C_28_H_42_O. This compound shows the following spectroscopic properties: The ^1^H NMR spectrum of 2 in CDCl_3_ showed signals attributable to an ABX-type aromatic ring at *δ_H_* 7.56 (d, *J* = 8.6 Hz, 2H), 7.38 (t, *J* = 2.2 Hz, 2H), 7.15 (dd, *J* = 8.6, 2.5 Hz, 2H), which indicated the presence of a 1,2,4-trisubstituted benzene ring. In addition, the locations of *δ_H_* 1.36 (s, 18H) and *δ_H_* 1.31 (s, 18H) suggest that there are two sets of non-equivalent hydrogen signals, each containing three methyl groups. ^13^C NMR data show that compound **2** has 12 carbon signals, including eight olefins or aromatic hydrocarbons (*δ_C_* 147.80, 147.76, 147.23, 138.66, 138.60, 124.61, 124.13, 119.24), two quaternary carbons (*δ_C_* 35.02, 34.67) and two obvious methyl groups (*δ_C_* 31.59, 30.33). After identification, the above data of ^1^H-NMR and ^13^C-NMR (Appendix A) are basically consistent with those reported in [40]. Thus, compound **2** was considered to be 1,1′-Oxybis(2,4-di-tert-butylbenzene) (Figure 2).

Glycerol monolinoleate (**3**): Compound **3** was extracted as a colorless oil, and its molecular formula is C_21_H_38_O_4_. This compound shows the following spectroscopic properties: The structural type of compound **3** was determined to be fatty acid from ^1^H NMR and ^13^C NMR data. ^13^C NMR data can clearly observe that compound **3** has a carbonyl (*δ_C_* 174.35) and two four olefin carbon signals (*δ_C_* 130.25, 130.02, 128.09, 127.91). It is inferred that *δ_C_* 70.27 is connected with carbonyl carbon and olefin carbon, and the residual carbon signals are mostly concentrated in the low chemical shift in the high field region, suggesting the existence of fat chains. The above NMR data (Appendix A) are basically consistent with those reported in the literature [41]. Thus, compound **3** was elucidated as glycerol monolinoleate (Figure 2).

Volemolide (17R)-17-methylincisterol (**4**): Compound **4** was extracted as a colorless oil, and its molecular formula is C_22_H_34_O_3_. This compound shows the following spectroscopic properties: ^13^C NMR data showed that compound **4** contains an obvious carbonyl carbon (*δ_C_* 170.70) and four olefinic carbons (*δ_C_* 169.11, 134.65, 132.87, 114.21). The combination of characteristic carbon signal and residual carbon signal shows that there is a tricyclic plus side chain in the structure of compound **4**, which is similar to that of sterols. The above NMR data (Appendix A) are basically consistent with those reported in the literature [42]. Hence, compound **4** was identified as volemolide (17R)-17-methylincisterol (Figure 2).

(24S)-ergosta-7-en-3-ol (**5**): Compound **5** was extracted as a colorless acicular crystal (dichloromethane), and its molecular formula is C_28_H_48_O. This compound shows the following spectroscopic properties: ^13^C NMR data show that compound **5** contains 28 carbon signals, in which two olefinic carbons at *δ_C_* 139.64, 117.43, hydroxyl carbon at *δ_C_* 71.09 and five methyl carbon at *δ_C_* 19.04, 17.61, 15.45, 13.05, 11.86 can be obviously observed. combined with all the ^1^H NMR and ^13^C NMR data, it can be inferred that the structural type of compound **5** is sterol. The above NMR data (Appendix A) are basically consistent with those of the literature [43]. Hence, compound **5** was identified as (24S)-ergosta-7-en-3-ol (Figure 2).

Dibutyl phthalate (**6**): Compound **6** was extracted as a colorless transparent oily liquid, and its molecular formula is C_16_H_22_O_4_. This compound shows the following spectroscopic properties: ^1^H NMR data showed that compound **6** is a typical ortho-substitution of benzene ring [*δ_H_* 7.71 (dd, *J* = 5.7,3.3 Hz, 2H), 7.52 (dd, *J* = 5.8,3.3 Hz, 2H)] and symmetrical [*δ_H_* 4.30 (t, *J* = 6.7 Hz, 4H), 1.75–1.68 (m, 4H), 1.48–1.40 (m, 4H), 0.96 (t, *J* = 7.4 Hz, 6H)]. ^13^C NMR data show that compound **6** has carbonyl (*δ_C_* 167.76), quaternary carbon (*δ_C_* 132.31), alkene carbon (*δ_C_* 130.94, 128.85), dioxycarbon (*δ_C_* 65.59), methylene carbon (*δ_C_* 30.57,19.19) and methyl carbon (*δ_C_* 13.74). The above NMR data (Appendix A) are basically consistent with those reported in the literature [10]. Thus, compound **6** was identified as dibutyl phthalate (Figure 2).

### 2.2. Antifungal Activity of Compounds **1**–**6**

*T. mentagrophytes* and *T. rubrum* are illustrative examples of non-yeast fungi, whereas *C. albicans* and *C. neoformans* belong to the yeast category. Consequently, an assessment of antifungal activity was conducted on these four types of fungi (see Figure 3). Notably, at a concentration of 25 μg/mL, compounds **1**, **3**, and **6** displayed inhibitory rates surpassing 50% against *T. mentagrophytes*, with compound **3** exhibiting the most potent activity. For *T. rubrum*, compounds **2** and **6** exhibited inhibitory rates exceeding 50%, with compound **6** demonstrating the most robust activity. The inhibitory rates of compounds **1**, **2**, **3**, and **4** against *C. albicans* ranged between 20% and 40%, with compound **4** exhibiting the most pronounced effect. However, all of these compounds exhibited relatively modest antibacterial activity against *C. neoformans*.

### 2.3. Discussion

*A. blazei*, a rare and valuable edible fungus that serves as both a nourishing food and a medicinal resource, is renowned for its remarkable nutritional benefits in everyday consumption. While its abundant nutritional value is commonly harnessed for dietary supplementation, comprehensive investigations into its bioactive constituents have been limited among scholars. Furthermore, there is a dearth of literature detailing the extraction and isolation of its chemical compounds, as well as a lack of in-depth exploration of its biological functionalities. Recent advancements in scientific and medical research have brought to light the potent and distinctive therapeutic attributes of naturally derived compounds. These encompass a spectrum of effects including antibacterial, anti-allergic, anti-cancer, immunosuppressive, and anti-inflammatory properties. Notably, the antifungal potential of the chemical constituents within *A. blazei* has remained largely unexplored until now.

In this study, compounds **1**–**6** were successfully isolated from *A. blazei*’s fruit bodies for the first time. After spectroscopic methods we can know that these six compounds were identified as linoleic acid (**1**), 1,1′-oxybis(2,4-di-tert-butylbenzene) (**2**), glycerol monolinoleate (**3**), volemolide (17R)-17-methylincisterol (**4**), (24s)-ergosta-7-en-3-ol (**5**), and dibutyl phthalate (**6**). The previous literature reports on *A. blazei* are mostly focused on anti-tumor, anti-virus, enhancing immune function and so on, but there are no related reports on antifungal activity. Therefore, in order to better understand and apply the resources of *A. blazei* and clarify its chemical composition and pharmacodynamic basis, this study was carried out. It is worth noting that all of these purified compounds were first found in *A. blazei*. The antifungal activities of these compounds were studied for the first time in vitro. The results showed that compounds **1–6** showed different sensitivities to different fungi. Particularly noteworthy was the exceptional inhibitory effect of compound **2** on *T. mentagrophology*, while compound **3** demonstrated a notable inhibitory effect against *T. rubrum*, and compound **6** displayed a high level of inhibitory activity against *C. albicans*. This comprehensive dataset underscores the medicinal significance of *A. Blazei* as a potent antifungal agent against pathogenic fungi. Moreover, it has the potential to reshape current perceptions surrounding *A. blazei* mushrooms and their analogous counterparts, which may dispel preconceived notions. The findings of this study are expected to stimulate further investigation from researchers, encouraging more profound explorations. Generally speaking, the above findings are helpful to promote the research progress of chemical constituents and antifungal activity of *A. blazei*. Nevertheless, it is pertinent to acknowledge that the present examination of individual compounds from *A. blazei* remains at a relatively preliminary stage, and this study has certain limitations. A future avenue of research necessitates the isolation of additional compounds, thereby facilitating more intricate analyses into the underlying mechanisms of action of *A. blazei* within an in vitro context.

However, it is essential to acknowledge that the present examination of individual compounds from *A. blazei* remains at a relatively preliminary stage. While this study marks a partial progress in our understanding of *A. blazei*’s medicinal properties, it also has certain limitations. To fully unlock the therapeutic potential of *A. blazei*, future research endeavors should focus on the isolation and characterization of additional compounds, facilitating more intricate analyses into the underlying mechanisms of action within an in vitro context. One promising avenue for further exploration is the elucidation of the synergistic effects of these compounds when combined. It is possible that the unique chemical composition of *A. blazei* contributes to its antifungal properties, and a more comprehensive study of these compounds in combination could provide valuable insights into their collective efficacy. In addition, research on the mechanism of antimicrobial action is crucial to help modify its structure, reduce toxicity and increase efficacy, and translate the research results into practical medical applications.

## 3. Materials and Methods

### 3.1. Reagents and Instruments

The *A. blazei* samples were sourced from an *A. blazei* cultivation base situated in Shiming Village, Si Qian Town, Jiangmen City, Guangdong Province. These samples were provided in the form of dried mushrooms. The extraction process utilized solvents of analytical grade, which were procured from Tianjin Yongda Chemical Reagent Co., Ltd. (Tianjin, China). These solvents encompassed methanol, petroleum ether, dichloromethane, ethyl acetate, and ethanol, among others. For chromatography, the essential reagents included methanol and ultrapure water, which were acquired from Fisher/Annergi Chemistry of Thermo Fisher Scientific (Shanghai, China).

Electric heating sleeve: Model 98-1-B from Tianjin Tester Instrument Co., Ltd. (Tianjin, China); electronic analytical balance: Maximum of 220 g, from Sedolis Co., Ltd.; semi-prepared liquid chromatography: Model LC-52 from Cypress (Beijing) Technology Co., Ltd. (Beijing, China); high-performance liquid chromatography: Agilent 1200 from Agilent Technologies; rotary evaporator: Model R-260PRO from BUCHI Labortechnik AG; rapid preparation liquid chromatography: Model Biotage Isolera One from Shanghai Musen Biotechnology (Shanghai, China); nuclear magnetic resonance spectrometer: Model AVANCE III 500 MHz/600 MHz from Bruker.

### 3.2. Separation and Purification

A total of 7.5 kg of dried *A. blazei* mushrooms was meticulously ground using a grinder and then packed into non-woven traditional Chinese medicine bags. A portion of the dried ABM mushroom powder was carefully placed into a sizeable flask and submerged in methanol. The flask, outfitted with an electric jacket and connected to a condenser, underwent reflux for roughly 2 h. Following this, the mixture was filtered, and a fresh solvent was introduced to the system, initiating a subsequent 2 h reflux. This iterative process was repeated until the complete extraction of the ABM dried mushroom samples was accomplished. The amassed extracts were pooled, and methanol was subsequently recuperated using a large-scale rotary evaporator to yield a concentrated ABM extract. The concentrated extract was then introduced into the rotating vessel of a sizable rotary evaporator, while ensuring that the quantity of the extract added at each instance did not surpass half of the capacity of the rotary bottle. Manipulating the water bath’s temperature and vacuum conditions, along with adjusting the rotation speed, facilitated the gradual addition of the ABM extract as the solvent within the rotating vessel as water evaporated. This cycle was repeated until all extracts were effectively concentrated, culminating in the recovery of the methanol solvent and the acquisition of the preliminary fraction of the crude methanol extract of *A. blazei*.

For column chromatography with silica gel, the ratio of ABM concentrate/mixed silica gel (80–100 mesh)/chromatographic silica gel (200–300 mesh) = 1:1:8. Petroleum ether extract (ABM-A), dichloromethane extract (ABM-DCM), ethyl acetate extract (ABM-B), acetone extract (ABM-BT), and methanol extract (ABM-E) were obtained via gradient elution with petroleum ether, dichloromethane, ethyl acetate, acetone, and methanol, respectively. Finally, these extracts were concentrated to obtain individual solvent concentrates, which were stored until use. The preliminary fractionation activity screening (secondary fraction) was carried out, and the same method was also used for the silica gel column chromatography.

For the ODS column chromatography, after analysis using TLC, HPLC, ELSD and other methods, the sample, mixed sample ODS, and chromatographic ODS were mixed at a ratio of 1:1:20 for each fraction, and then loaded into the appropriate column. Rough cut: the proportion of methanol to water was subjected to gradient elution at 30%, 50%, 70%, and 100%, respectively, and then collected into different fractions. Then, the TLC point plate, HPLC liquid phase, and evaporative light-scattering detector (ELSD) were used again for spectrum analysis and quality analysis.

Acetone crude extract and dichloromethane crude extract were separated and purified using a combination of silica gel column chromatography (sample/mixed sample silica gel/chromatography silica gel ≈ 1:1:8), ODS column chromatography (sample/mixed sample ODS/ODS = 1:1:20), and Biotage HPLC (methanol/water). The obtained compounds are listed below.

ABM-DCM (20 g) was eluted using silica gel column chromatography (PE/EA) to obtain 14 segments of small-fraction ABM-DCM-A~N. ABM -DCM-F (50 mg) was directly synthesized using HPLC (96% Me, flow rate of 3 mL/min) to obtain compound **2** (16.8 mg). ABM-DCM-L (ABM-DCM was eluted using silica gel column chromatography, with PE/EA = 100: 15 cm 100: 50, 3.15 g) was eluted with methanol/water using ODS chromatography to obtain nine segments, of which ABM-DCM-L-6 (312 mg) was semi-prepared using HPLC (Cyprus, Phenomenex semi-preparation column, 82% Me, flow rate of 5 mL/min) to obtain compound **3** (22.7 mg) and compound **4** (4.3 mg). Compound **5** (104.2 mg) was prepared from small-fraction ABM-DCM-J-93%-4 (≈220 mg) via conditional touch and HPLC (Cypress, phenomenex semi-preparation column, 92% Me, flow rate of 5 mL/min). Compound **6** (220 mg) was directly obtained via ODS chromatography with methanol/water (25–100%).

### 3.3. Antibacterial Activity Determination

In this study, four fungal strains, *T. mentagroph*, *T. rubrum*, *C. albicans*, and *C. neoformans*, were cultured in vitro.

Culture preparation: for the aseptic operation, 2.5 mL of the medium was added to a 15 mL fast cap culture tube, a single colony was carefully removed from the Petri dish using a sterile rod, and the tip of the rod was rotated in the culture medium (while being careful not to touch the end of the rod at anything, except the colony and the medium); the culture tube was placed in a 37 °C incubator, at a 250 rpm speed oscillation, for culturing to the logarithmic phase.

Culture plate preparation: compounds **1**–**6** at 0.2 mg each were dissolved in 200 μL of DMSO, and 5 μL of the sample solution and DMSO as a control were added to a 96-well plate; then, the culture plate was put into reserve.

Diluted pathogen culture: 5 mL of aseptic medium and 500 μL of culture were added to a 15 mL conical tube and mixed, and then the medium or culture was used as diluent to dilute OD600 to 0.030–0.060. After mixing the diluted culture with the culture medium at a proportion of 1:10, 195 μL of inoculation medium and 200 μL of aseptic culture medium were added to a 96-well plate as the blank group, and then the plate was placed in a sealed bag. The plate was fixed in a vibrating screen with a plate bracket or tape and cultured at 37 °C for 16 h.

### 3.4. Statistical Analysis

All in vitro assays were performed in triplicate. The data were analyzed and processed using GraphPad Prism 9.0 software. The data were expressed as mean ± SD. Differences between the groups were compared using a *t*-test. *p*-value < 0.05 was considered as representing a statistically significant difference.

## 4. Conclusions

In recent years, the exploration of *A. blazei*, a rare and valuable edible fungus, has garnered increased attention due to its dual role as a nourishing food and a potential medicinal resource. Its reputation for remarkable nutritional benefits in everyday consumption has made it a sought-after dietary supplement. However, despite its extensive use, comprehensive investigations into its bioactive constituents and their potential medicinal properties have remained limited among scholars. Furthermore, there is a notable dearth of literature detailing the extraction and isolation of its chemical compounds, as well as a lack of in-depth exploration of its biological functionalities.

In this study, compounds **1–6** have been successfully isolated from the non-polysaccharide constituents of *A. blazei* via processes like column chromatography and semi-preparative HPLC. These compounds underwent meticulous characterization via techniques such as ^1^H-NMR and ^13^C-NMR spectroscopy. These six compounds have been identified as linoleic acid (**1**), 1,1′-oxybis(2,4-di-tert-butylbenzene) (**2**), glycerol monolinoleate (**3**), volemolide (17R)-17-methylincisterol (**4**), (24s)-ergosta-7-en-3-ol (**5**), and dibutyl phthalate (**6**). Of particular significance is the fact that all these purified compounds were previously unidentified in *A. blazei*. Additionally, a pioneering assessment of the in vitro antifungal activities of these compounds against *T. mentagrophytes*, *T. rubrum*, *C. albicans*, and *C. neoformans* was carried out for the first time. The results of this assessment unveiled varying degrees of sensitivity against the four fungi, with specific compounds exhibiting distinct inhibitory effects. Compound **2** emerged as particularly effective against *T. mentagrophytes*, while compound **3** demonstrated a notable inhibitory effect against *T. rubrum*. Remarkably, compound **6** displayed a high level of inhibitory activity against *C. albicans*. Different types of compounds in *A. blazei* have been shown to have different antifungal activities against different fungi. The identified compounds have simple structures, including fatty acid glycerides and aromatic compounds, and have the potential to be developed into novel antifungal agents. In a word, *A. blazei*, as a representative macrofungus, demonstrates inherent antifungal activities. This study informs for further research and advancements related to *A. blazei*, positioning it as a potential source of natural antifungal products. Moreover, it contributes to expanding the knowledge on the structural diversity of natural compounds and facilitating the elucidation of *A. blazei*’s effective material basis.

However, it is essential to acknowledge that the present examination of individual compounds from *A. blazei* remains at a relatively preliminary stage. While this study represents a significant leap forward in our understanding of *A. blazei*’s medicinal properties, it also has certain limitations. To fully unlock the therapeutic potential of *A. blazei*, future research endeavors should focus on the isolation and characterization of additional compounds, facilitating more intricate analyses into the underlying mechanisms of action within an in vitro context. One promising avenue for further exploration is the elucidation of the synergistic effects of these compounds when combined. It is possible that the unique chemical composition of *A. blazei* contributes to its antifungal properties, and a more comprehensive study of these compounds in combination could provide valuable insights into their collective efficacy. Moreover, in vivo studies are essential to validate the in vitro findings and to assess the safety and efficacy of *A. blazei* as a potential antifungal therapy in living organisms. Clinical trials involving *A. blazei*-based treatments may be the next step in translating these promising findings into practical medical applications.

In conclusion, the discovery of compounds with antifungal properties within *A. blazei* marks further progress in our understanding of this remarkable fungus’s medicinal potential. While there are still many questions to be answered and challenges to be overcome, it deserves further study.

## Figures and Tables

**Figure 1 molecules-28-07321-f001:**
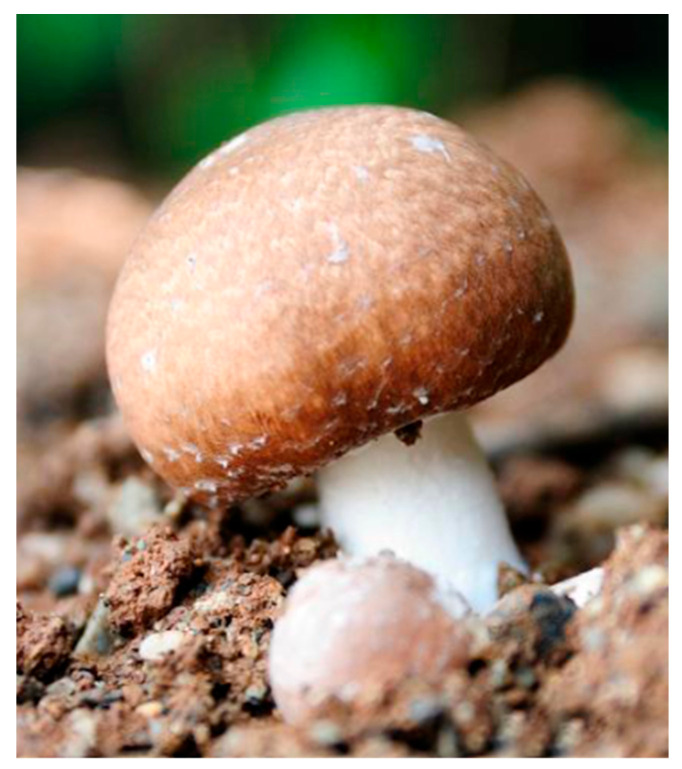
The fruit body of *Agaricus blazei* Murrill.

**Figure 2 molecules-28-07321-f002:**
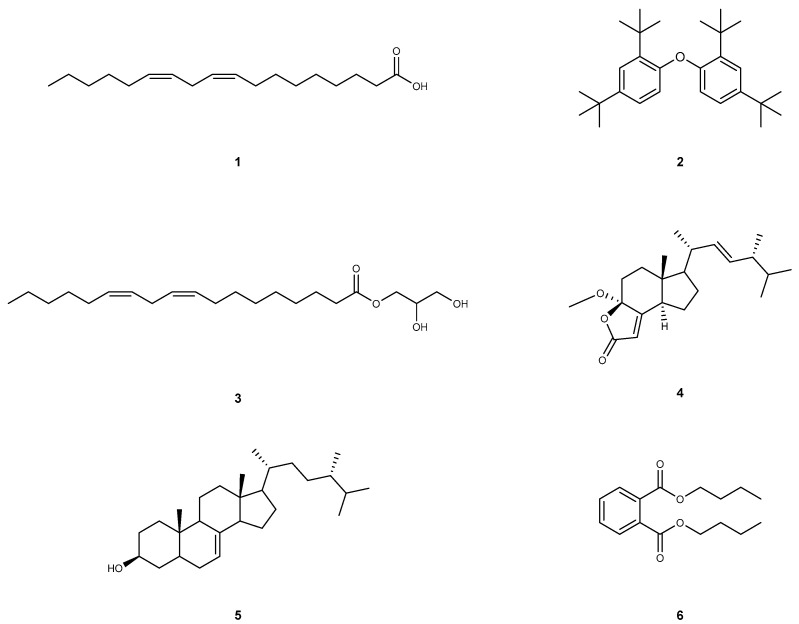
The structure of compounds **1**–**6**.

**Figure 3 molecules-28-07321-f003:**
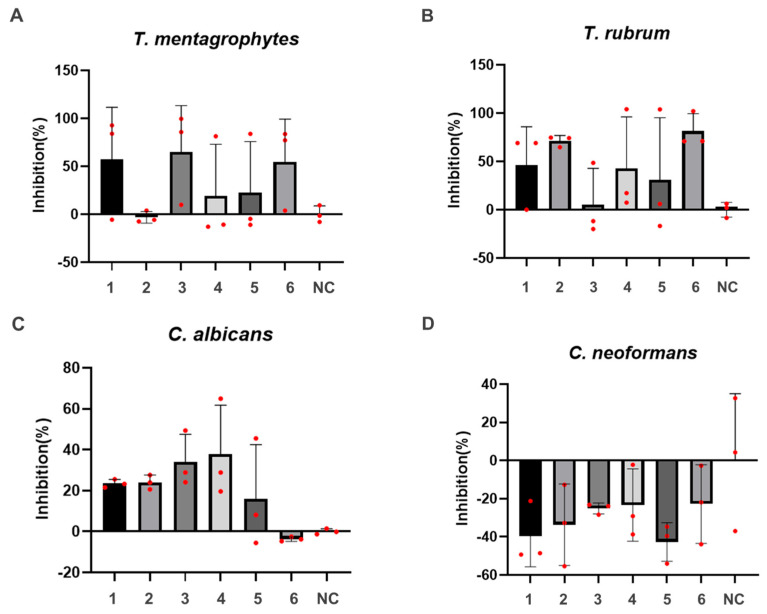
Effects of compounds **1**–**6** on four fungi. (**A**) *T. mentagroph*, (**B**) *T. rubrum*, (**C**) *C. albicans,* (**D**) *C. neoformans*.

## Data Availability

Not applicable.

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
