# Peer review of "Isolation and Identification of Chemical Compounds from Agaricus blazei Murrill and Their In Vitro Antifungal Activities"

_molecules, 2023, doi:10.3390/molecules28217321_

Round 1

Reviewer 1 Report

Comments and Suggestions for Authors

Dear authors

Concerning the manuscript entitled “Isolation and identification of chemical components from Agaricus blazei Murrill and their antifungal activities in vitro” which was submitted to the molecules journal by Rui-Rui Yu et al., there are some comments for the manuscript improvement included as followings:

Title:

The title part is appropriate

Abstract

The title part is appropriate

Introduction

The introduction section is appropriate. Why the end of the introduction is exactly the same as the 2-    Abstract  of the article,  There is no need to give the all result in the introduction, the authors explain about it in abstract.

Results

Subscripts should be used to write chemical formulas. Correct all for example: C18H32O2 change to C18H32O2

Manuscript body sections

This part is suitable

But , Part 3.3. the author wrote Antibacterial activity determination:

Antibacterial or antifungal? Please correct it .

Conclusion

It is suitable, however inclusion of main drugs examples in the conclusion will make the findings of the manuscript more appropriate and understandable.

References

This papers can be suitable

A recent overview of 1, 2, 3-triazole-containing hybrids as novel antifungal agents: focusing on synthesis, mechanism of action, and structure-activity relationship (SAR)

Novel 1, 2, 4-Triazoles as Antifungal Agents

With best regards

Comments on the Quality of English Language

Extensive editing of English language required

Reviewer 2 Report

Comments and Suggestions for Authors

This work is devoted to the isolation and identification of the chemical components of Agaricus blazei Murrill. The authors also used these substances to study their antifungal activity in vitro. In general, this article is written in an understandable and accessible language, the main ideas are not in doubt. The results obtained by the authors are confirmed by modern methods. In terms of volume, structure and content, this work meets the requirements of the Journal. There are some points that would like to be improved:

1. Abstract is desirable to expand. Now it looks quite modest.

2. The introduction can also be expanded. It would be desirable to tell in more detail about the problems and the topic of this study.

3. It is desirable to double-check the text. The text often contains "Error! Reference source not found".

4. It is desirable to add other methods of component analysis.

5. It is desirable to compare the results obtained in more detail with the data from the literature.

6. Please cite: 10.3390/molecules27186129.

7. In general, this work makes a good impression and can be recommended for publication after some revision.

Reviewer 3 Report

Comments and Suggestions for Authors

It is not necessary to repeat Murril after the first denomination of the plant.

Lines 36 and 39 change Agaricus Blazei and A. Blazei in Agaricus blazei and A. blazei and later in all the MN

Line 39 change Baicao Classic" an ancient in Baicao Classic", an ancient and add a reference

Page 30 change the genus A. Blazei in the A. blazei, in any case, A blazei is a species and not a genus, and therefore it is not clear if the sentence is about the genus or the species. Later it is reported as species in the Conclusions. It is should be reported a careful information if the isolated compounds are present in other plants.

Line 53 change T. mentagrophytes, T. rubrum, C. albicans, and C. neoformans in vitro in T. mentagrophytes, T. rubrum, C. albicans, and C. neoformans in vitro

Line 62 change C18H32O2 in C18H32O2

2.1. Identification of Compounds 1–6. Being the identified compounds already known and the spectroscopic data already reported, just report the necessary references.

Line 111 change 25ug/ml in 25 ug/mL

Figure 2 change (T.mentagroph, T.rubrum, C.albicans, C.neoformans). in (T. mentagroph, T. rubrum, C. albicans, C. neoformans).

Line 134 in the sentence These six compounds included change in These six compounds are

Line 138 change in vitro in in vitro

Line 140 check the sentence

All the phytochemical part could be reported in few lines.

The general consideration is that the research do not have relevant novelty and could be better reported as Short Communication, limiting the information to the essential.

Comments on the Quality of English Language

There are many different errors in The MN

Round 2

Reviewer 3 Report

Comments and Suggestions for Authors

The Introduction is too long and plenty of information not closely related to the research, including consideration about China! Information should be complementary to the research and evidently related to the aims of the paper, otherwise the MN increases its pages without real value.

Page 2 line 69 the name of the Agaricum species is completely wrong

Line 87 it is not a genus but a species! Please restudy Biology

The last part of the Introduction contains results, this is not in accordance with the aims of an Introduction

RESULTS

All the reported constituents are already known, the analytic data must not be reported

Idem for the figure absolutely not necessary, pages must be utilized in the adequate way, because they are precious

No any information about the relative percentages, which on the contrary are relevant

Page 4 lines 156-157 the best classification is yeast category!!!

Section 3.2 is full of details without any relevance whereas other basic information is lacking. Again, a wasting pages

Conclusion must be totally revised for several reasons, but in particular the sentence “this study opens the door to a new area of research” is incredible!

These are some of the comments, with other ones, supporting the Reject of the paper.

Comments on the Quality of English Language

all the MN needs revision

Round 3

Reviewer 3 Report

Comments and Suggestions for Authors

I noticed that still the NMR data and assignments are present in the MN. Considering that all the constituents are already known, the usual writing is not report the data and only refer to the already published data.
